# Investigation of Sliding Mode Control in the Nonlinear Modeling of Cordless Jigsaws

**DOI:** 10.3390/s25020456

**Published:** 2025-01-14

**Authors:** Sándor Apáti, György Hegedűs, Sándor Hajdu, Péter Korondi

**Affiliations:** 1Department of Electrical Engineering and Mechatronics, Faculty of Engineering, Vehicles and Mechatronics Institute, University of Debrecen, Ótemető Str. 2-4., H-4028 Debrecen, Hungary; hajdusandor@eng.unideb.hu (S.H.); korondi.peter@eng.unideb.hu (P.K.); 2Faculty of Mechanical Engineering and Informatics, Institute of Machine Tools and Mechatronics, University of Miskolc, Miskolc-Egyetemváros, H-3515 Miskolc, Hungary; gyorgy.hegedus@uni-miskolc.hu

**Keywords:** cordless jigsaw, sliding mode control, electromechanical model, MATLAB/simulink

## Abstract

The aim of this paper was to reduce the current spikes in battery-powered saw motors by designing and implementing a sliding mode model-following adaptive controller. The proposed controller reduces overcurrent consumption, improves system energy efficiency, and effectively maximizes battery runtime, especially under high-load conditions. By applying nonlinear compensation techniques, the controller can ensure smooth motor operation, reduce mechanical stress, and prolong tool life. The results showed that this control strategy is particularly suitable for hand tools, where a long battery life is essential for efficient operation.

## 1. Introduction

Hand tools are indispensable in machining and production processes. The most versatile of these are cordless jigsaws, which can cut a wide range of materials, both straight and curved, and soft and hard. The cordless design offers many advantages, such as mobility and independence from electrical connections, which is particularly useful in remote and hard-to-reach areas. The battery-powered setup ensures reliable cutting results and long periods of continuous use, even in DIY (do it yourself) environments, and new developments in electronics have made it easier to test and develop tools such as cordless jigsaws. Their cyclic and bursty operating characteristics pose challenges for standard test and inspection solutions, so these devices require specialized procedures tailored to their unique dynamics [1,2].

Advanced control techniques such as sliding modal control (SMC) have become popular in response to these challenges; SMC is a well-known technique for controlling uncertainty and disturbances in control systems, due to its flexibility and ability to manage nonlinear dynamics. Reviewing the theoretical underpinnings of SMC is crucial to maximizing its potential. The idea of variable structure systems, created in the Soviet Union in the 1950s and 1960s, is the basis of SMC. Vadim I. Utkin [3] enabled the broad use of this theory in a variety of fields, including robotics [4], servo drives [5], aviation, and, latterly, induction motor drives [6]. This characteristic has made SMC one of the essential components of contemporary control systems, particularly in intricate, nonlinear applications [7]. One of the biggest challenges in implementing SMC is “chatter”, which is a high-frequency jitter brought on by abrupt changes in the control input [8].

The theoretical features of SMC, particularly the requirements for stability and the function of feedback dynamics in attaining desirable system responses, were established by early foundational works, such as that of Sira-Ramirez (1993) [9]. Building on these ideas, real-world applications of SMC have been investigated in very precise systems, including robot manipulators. These studies highlighted how crucial it is to create reliable control systems, while maximizing the tracking accuracy and energy efficiency [10].

Through the resolution of particular technical issues, recent advancements have considerably broadened the scope of SMC. Discrete-time sliding mode control techniques, for example, have been shown to be successful in minimizing “chattering”, especially in digital systems where the exact timing of control intervals is essential [11]. A potent method for improving the position tracking and attitude adjustment in mechanical systems, such as drilling tools, is observer-based adaptive sliding mode control, which incorporates state observers. This illustrates SMC’s adaptability in addressing real-world issues, while preserving exact control [12].

Customizing SMC approaches to manage intricate nonlinear systems and fluctuating dynamic situations has been the main focus of recent advancements. These methods are especially applicable to energy-efficient gadgets and autonomous systems, where resilience and flexibility are essential. Additionally, advancements like sliding mode observers with fixed-time convergence have greatly enhanced state monitoring and quick reaction times, making them extremely useful for dependability and fault diagnosis enhancement [13,14].

Notwithstanding these developments, nothing is known about the use of SMC in battery-operated, portable instruments like cordless jigsaws. Innovative control techniques are required to address the particular difficulties posed by cordless tools, including their nonlinear and periodic mechanical motion, abrupt current spikes, and the constraints of battery-powered designs. Furthermore, the inherent high-frequency switching of SMC can exacerbate “chatter” and make its use in these devices more difficult. These issues need to be addressed to optimize the performance, energy savings, and operational reliability of cordless tools.

Recent developments in SMC highlight its importance in nonlinear dynamic systems and its suitability for resolving contemporary engineering problems, in order to support this environment. With an emphasis on enhancing transient performance and shock resilience, a number of research studies have investigated the efficacy of nonlinear feedback control systems [15]. These results show that SMC can maintain accuracy and durability in a variety of settings, which makes it a workable solution to the particular problems of cordless jigsaws. These results are expanded in this study by developing a novel dual-observer paradigm that is appropriate for the unique dynamics of wireless devices. The suggested architecture combines a first-order reference model with a low-order state-space observer to manage unmodeled dynamics, enhance system stability, and offer precise state monitoring.

The integration of this dual-observer system has several advantages, including robust angular rate control, minimized “chattering” effects, and efficient energy consumption. By minimizing the outages brought on by the initial current peaks and loads, the suggested method guarantees steady and effective operation, enhancing SMC theory and practice. One of the primary issues with SMC, “chattering”, may be reduced with the use of discrete-time sliding mode control. The “chattering” from high-frequency switching in SMC makes it challenging to attain reliable and efficient control. For digital systems, where the proper control range must be identified to lessen this effect and boost system robustness, a discrete-time approach is especially crucial [11].

These developments have real-world uses in nonlinear control systems, such as multirotor UAV stabilization. These solutions show how SMC can decrease outages and boost flexibility in dynamic contexts. By tackling the following problems, SMC technology demonstrates its suitability for contemporary engineering applications [16].

One of the major advancements in SMC is the employment of sliding mode observers with constant-time convergence to guarantee precise condition monitoring and quick control reaction times. When precision and dependability are needed, these observers work well in mechanical systems [13]. In fault diagnostics, reduced-order sliding mode observers have also gained popularity, contributing to the increased dependability and efficiency of control systems across a range of industries [14].

These developments are essential for the creation of nonlinear control systems, particularly in applications involving battery-powered machinery, where dependability, stability, and energy efficiency are vital. Notwithstanding these successes, there are unique difficulties in integrating SMC into portable, battery-operated devices like cordless jigsaws. Inrush current spikes are frequently caused by the periodic, nonlinear mechanical motion of such equipment, underscoring the drawbacks of wireless systems, including issues with energy efficiency and battery life. Furthermore, SMC is particularly vulnerable to the small-scale dynamics of such equipment, because of the intrinsic high-frequency switching that intensifies “chattering” effects.

To address these problems, there is a growing need for advanced control techniques that enhance the performance and efficiency of portable tools, while simultaneously extending their operational lives. Prior research has examined chip formation dynamics in machining processes [17,18] and cutting forces and tool wear in forestry and wood harvesting [19,20,21,22], but little emphasis has been paid to optimizing cordless jigsaws. Improving energy efficiency, reducing current spikes, and ensuring consistent performance under different load circumstances are understudied topics. In order to bridge this gap, this study used sliding mode control techniques to address these specific issues and enhance the tool’s electrical and mechanical performance.

SMC has grown to be an essential part of power electronics and motion control over the years, because of its ability to effectively handle nonlinearities and uncertainties [23,24].

Since the sliding surface design directly affects the trajectory’s rate of convergence, it is an essential part of SMC. The slope of the sliding surface is a crucial component in finding a balance between durability and speed. Alternative approaches, such as altering the τ coefficient constantly or intermittently, have been investigated to overcome this [25]. However, problems still exist, especially when it comes to controlling high-frequency switching, which might resonate with the inherent frequency of a system and result in oscillations. To reduce these oscillations and guarantee stability and dependable operation, model-based disturbance estimators have been presented [26].

Although sliding mode control has been extensively researched in areas such as robotics and induction motor drives, the use of sliding mode control in portable, battery-powered equipment is still poorly understood. Cordless jigsaws and other portable tools present special challenges that require specialized solutions to deal with their constraints and operating environment. Unlike large systems and continuous industrial machinery, cordless equipment requires specialized control techniques to manage initial current spikes, load-induced outages, and energy efficiency issues. Given these challenges, this work attempts to bridge the gap between the theoretical development of sliding mode control and its actual use in battery powered equipment. This method lowers the cordless jigsaw’s first current peak, while optimizing the energy efficiency and operating stability. By combining a first-order reference model with a low-order state-space observer, the suggested dual-observer structure provides a comprehensive method to tackle the difficulties presented by unmodeled dynamics. This framework tackles significant problems, including chatter, which lowers the control precision and shortens the lifespan of machine components, in addition to enhancing the control system’s dependability by guaranteeing precise monitoring of state variables. There are several noteworthy operational benefits of integrating a dual-observer system. It guarantees precise and seamless operation under constantly fluctuating load circumstances and offers strong angular speed control. Additionally, by reducing the loosening effects typical of sliding mode control, the frame increases the control precision and lessens the wear on machine components. Additionally, the dual-observer system maximizes energy consumption to prolong battery life and boost cordless tool efficiency. When combined, these advancements can theoretically and practically help to guarantee the dependability and effectiveness of cordless jigsaws in real-world scenarios.

To further enhance system responsiveness, relay-based control rules are incorporated into the control structure. This method is particularly suitable for circumstances requiring prompt action, since it offers a straightforward analog solution that can attain a high switching frequency. The inherent limitation of relay-based methods in ensuring sliding mode formation over the whole state space is addressed using pulse-width modulation (PWM). PWM effectively overcomes these constraints on mechanical systems, particularly when the principal frequency of motion is less than 100Hz. In this study, the relay switching frequency is set to around 3kHz, ensuring constant and dependable system performance.

By integrating a dual-observer structure with relay-based control rules and PWM, the suggested system strikes a balance between accuracy, robustness, and energy economy. With these enhancements, the control system can react to dynamic shifts in load circumstances and maintain stable operation. This all-encompassing approach tackles the unique difficulties presented by cordless jigsaws and establishes the groundwork for further advancements in the sliding mode control of battery-operated devices.

This paper is organized as follows. The experimental setup utilized for validation is described in Section 5. The technical difficulties this study tackled are described in Section 2. The theoretical underpinnings of sliding mode control are explained in Section 3.1. The system’s equations are derived in Section 2.2, and the cordless jigsaw model is subjected to these theories in Section 4.1. The sliding mode control stability conditions are covered in Section 4.2. Simulation results with and without sliding mode correction are reported in Section 6. Insights into the prospects for further advancement in this discipline are provided in the last part, which summarizes the conclusions and future directions.

## 2. Problem Statement

### 2.1. Engineering Problem

During the startup of DC motors, a sudden surge in current often occurs, resulting in a high peak starting current. This phenomenon can negatively affect both the battery’s safe operation and lifespan, as well as the motor’s winding, as large current surges cause increased wear and heating. Nowadays, there is a growing effort to replace mechanical components with digital control systems, which have become more affordable thanks to advancements in the electronics industry. This approach makes it possible, for example, to eliminate mechanical transmissions, thereby simplifying the motor’s drivetrain and reducing maintenance requirements. However, the removal of transmissions also introduces challenges, as load-induced disturbances have a more direct impact on the motor, potentially causing instability or reduced performance.

The robustness and capacity of sliding mode control (SMC) to manage nonlinearities in control systems are well known. But there are particular difficulties when using battery-operated, portable tools like cordless jigsaws. These include the tool’s periodic and nonlinear mechanical motion, which causes abrupt current spikes, and the constraints of the cordless design, which is crucial for energy efficiency and battery life. Furthermore, SMC is particularly sensitive to the small-scale dynamics of such devices, due to the high-frequency switching that is inherent in the device, which can intensify “chattering” effects.

Applying appropriate control strategies, such as sliding mode control, is required to address these issues. Sliding mode control offers the ability to quickly and accurately modify the motor’s dynamic response to changing load conditions. By doing this, it guarantees steady operation and shields the motor’s parts from undue wear and tear. Sliding mode control is used in this study to reduce startup current peaks, lessen the impact of load variations, and improve the general dependability and performance of cordless jigsaws.

Based on a literature review, sliding mode control (SMC) is an effective tool for addressing challenges in nonlinear systems, such as current spikes and instability. This study builds on SMC by integrating a dual-observer approach to provide a comprehensive solution for reducing startup current spikes, mitigating the effects of load disturbances, improving system stability, and enhancing energy efficiency. Fast and accurate dynamic reactions are made possible by SMC, especially during starting, when changes in current have a direct effect on the battery and motor windings’ lifespan. By anticipating and compensating for these disruptions, the dual observer protects the battery and motor. Furthermore, removing mechanical gearboxes simplifies the powertrain but increases the motor’s vulnerability to load-induced disruptions. The dual observer detects and addresses these issues accurately, while SMC ensures stable operation under varying conditions. The flexibility and high-frequency switching of SMC allow optimizing the motor’s dynamic responses, which not only improves the stability but also reduces the energy consumption—especially critical for cordless jigsaws, where battery life and energy efficiency are paramount. This study builds on the theoretical foundations of SMC, such as the works of [3,27], while further advancing the discrete sliding mode control techniques introduced by [26] through the integration of a dual observer. It also complements the disturbance estimation methods described by [28] by developing a more robust model. Grounded in the literature, this research offers a practical and robust control strategy that minimizes current spikes, ensures stability, and enhances energy efficiency for battery-powered tools.

### 2.2. Electromechanical Model of the Jigsaw

Figure 1 illustrates an electromechanical model of a jigsaw. On the left side of the figure, a diagram of a permanent magnet DC motor is shown, which connects to a Scotch yoke mechanism through a gear connection. This mechanism ensures the reciprocating movement of the saw blade according to the following equation:(1)$$x\left(t\right)=e_{k}\left(1-\cos\varphi_{k}\right),$$
where x(t) is the blade position, and $e_{k}$, $\varphi_{k}$ are the eccentric angle of the Scotch yoke mechanism.

The relationship between the motor shaft angle and the gear angle of the Scotch yoke mechanism is determined by the $i_{mk}$ transmission ratio:(2)$$\varphi_{m}=i_{mk}\varphi_{k},$$
where $\varphi_{m}$ is the motor shaft angle. The time derivative of (Equation 2) gives the speed function of the saw:(3)$$\dot{x}\left(t\right)=e_{k}{\dot{\varphi }}_{k}\sin\varphi_{k},$$
where ${\dot{\varphi }}_{k}$ is the angular velocity of the Scotch yoke mechanism. The mathematical model of the jigsaw is derived using energy-based formulation, relying on displacement and charge variables. For this, the $L_{f}$ Lagrangian function, which in this case includes only the kinetic co-energy $T^{*}$ of the system and the magnetic co-energy $W_{m}^{*}$ of the coil, must be formulated based on the conservative components of the system. Based on Figure 1, the Lagrangian function $L_{f}$ for the conservative components can be written as(4)$$L_{f}=T^{*}+W_{m}^{*}=\frac{1}{2}J_{m}{\dot{\varphi }}_{m}^{2}+\frac{1}{2}J_{k}{\dot{\varphi }}_{k}^{2}+\frac{1}{2}m_{a}{\dot{x}}^{2}+\frac{1}{2}L_{a}{\dot{q}}^{2},$$
where $J_{m}$ is the moment of inertia of the motor shaft, $J_{k}$ is the moment of inertia of the gear, *m* is the mass of the reciprocating part of the Scotch yoke mechanism, $L_{a}$ is the inductance of the coil, and $\dot{q}$ is the time derivative of the charge, representing the current. The $k_{m_{k}}$ denotes the reciprocal of the transmission ratio between the motor and the eccentric shaft, $k_{m_{k}}=\frac{1}{i_{mk}}$. Substituting (Equation 2) into (Equation 4), we reduce the number of variables by one:(5)$$L_{f}=\frac{1}{2}\frac{J_{eq}\left(\varphi_{m}\right)}{J_{m}+J_{k}k_{m_{k}}^{2}+m_{a}e_{k}^{2}k_{m_{k}}^{2}\sin^{2}\left(k_{m_{k}}\varphi_{m}\right)}{\dot{\varphi }}_{m}^{2}+\frac{1}{2}L_{a}{\dot{q}}^{2}.$$
Using the notation $J_{eq}\left(\varphi_{m}\right)$, we obtain the equivalent moment of inertia of the system:(6)$$J_{eq}\left(\varphi_{m}\right)=\left[J_{m}+J_{k}k_{m_{k}}^{2}+m_{a}e_{k}^{2}k_{m_{k}}^{2}\sin\left(k_{m_{k}}\varphi_{m}\right)\right]$$
Thus, Equation (Equation 5) takes the following form:(7)$$L_{f}=\frac{1}{2}J_{eq}\left(\varphi_{m}\right){\dot{\varphi }}_{m}^{2}+\frac{1}{2}L_{a}{\dot{q}}^{2}$$
For the non-conservative components of the system shown in Figure 1, the virtual work can be formulated as follows: (8)$$\delta W_{nc}=U_{0}\delta q-\left(R_{a}+R_{b}\right)\dot{q}\delta q-k_{e}{\dot{\varphi }}_{m}\delta q+k_{m}\dot{q}\delta \varphi_{m}-F_{c}e_{k}k_{m_{k}}\sin\left(k_{m_{k}}\varphi_{m}\right)\delta \varphi_{m}-b{\dot{\varphi }}_{m}\delta \varphi_{m},$$
where $\delta W_{nc}$ represents the virtual work of the non-conservative components, $U_{0}$ is the terminal voltage, δq is the virtual change in charge, $R_{a}$ is the circuit resistance, $R_{b}$ is the battery resistance, $k_{e}$ is the motor electrical constant, $\delta \varphi_{m}$ is the virtual change in angle, $F_{c}$ is the cutting force, and *b* is the internal damping coefficient of the motor. The saw only cuts during the upward movement, while a blade-tilting mechanism prevents cutting during the downward movement. Thus,(9)$$F_{c}=\left\{\begin{array}{c} F_{c}\left(\varphi_{k}\right)\;\;\;\;\mathrm{if}\;\sin\left(\varphi_{k}\right)\;>\;0,\\ 0\;\;\;\;\;\;\;\;\;\;\mathrm{if}\;\sin(\varphi_{k})\;\leq \;0. \end{array}\right.\;$$
Given the Lagrangian function and the virtual work for the coordinates *q* and $\varphi_{m}$, the differential equations for the mathematical model can be derived from the Lagrange equations:(10)$$\frac{d}{dt}\left(\frac{\partial L_{f}}{\partial \dot{q}}\right)-\frac{\partial L_{f}}{\partial q}=-\left(R_{a}+R_{b}\right)\dot{q}-k_{e}{\dot{\varphi }}_{m},$$(11)$$\frac{d}{dt}\left(\frac{\partial L_{f}}{\partial \dot{x}}\right)-\frac{\partial L_{f}}{\partial x}=k_{m}\dot{q}-\left[F_{c}e_{k}k_{m_{k}}\sin\left(k_{m_{k}}\varphi_{m}\right)\right]-b{\dot{\varphi }}_{m}.$$
Based on (Equation 7) and (Equation 8), the differential equations of the electromechanical model can be obtained with the help of Lagrange equations:(12)$$L_{a}\ddot{q}+\left(R_{a}+R_{b}\right)\dot{q}+k_{e}{\dot{\varphi }}_{m}=U_{0},$$(13)$$J_{eq}\left(\varphi_{m}\right){\ddot{\varphi }}_{m}+\frac{1}{2}J_{eq}^{\prime }\left(\varphi_{m}\right){\dot{\varphi }}_{m}^{2}=k_{m}\dot{q}-F_{c}e_{k}k_{m_{k}}\sin\left(k_{m_{k}}\varphi_{m}\right)-b{\dot{\varphi }}_{m}.$$
Since the coefficients are not constants, the mechanical equation is extremely nonlinear, and ${\dot{\varphi }}_{m}$ seems squared. In addition, the cutting force depends on $\varphi_{m}$.

## 3. Methodology

This study employs a dual-observer structure that integrates a first-order reference model with a reduced-order state-space observer inside a sliding mode control framework. This innovative method instantly solves important problems, including lowering early current spikes, minimizing load-induced disturbances, and ensuring stable and efficient cordless jigsaw operation. According to current research in nonlinear feedback control, it is essential to identify and correct for disturbances in order to ensure a robust control system design [15]. The dual-observer structure presented in this study was developed in response to these findings, providing a cordless jigsaw with precise state monitoring and steady operation, even when dynamic load variations and unmodeled dynamics are present.

The suggested methodology is based on the theoretical underpinnings of observer-based sliding mode control techniques and model-referenced sliding mode control. The next subsections provide a detailed description of these methods.

### 3.1. Theoretical Basics of Model-Referenced Sliding Mode Control

To address the problems listed in Section 2.1, a model-referenced sliding mode control approach was applied. This section summarizes the essential background needed to understand the article, based on [29]. Consider a partially perturbed system:(14)$$\frac{d}{dt}\left[\begin{array}{c} x_{1}\\ x_{2} \end{array}\right]=\left[\begin{array}{cc} A_{11} & A_{12}\\ {\hat{A}}_{21}+∆A_{21} & {\hat{A}}_{22}+∆A_{22} \end{array}\right]\left[\begin{array}{c} x_{1}\\ x_{2} \end{array}\right]+\left[\begin{array}{c} B_{1}\\ {\hat{B}}_{2}+∆B_{2} \end{array}\right]u,$$
where $x_{1}\in R^{n_{1}}$ represents the unperturbed, and $x_{2}\in R^{n_{2}}$ represents the perturbed state variables vector. An observer, which is called a second-order reference model, is constructed for the perturbed state variable, with parameters $A_{11}\in R^{n_{1}\times n_{1}}$, ${\hat{A}}_{22}\in R^{n_{2}\times n_{2}}$, $A_{12}\in R^{n_{1}\times n_{2}}$, ${\hat{A}}_{21}\in R^{n_{2}\times n_{1}}$, $B_{1}\in R^{n_{1}\times n_{2}}$, ${\hat{B}}_{2}\in R^{n_{2}\times n_{2}}$, and $u\in R^{n_{2}}$ [30,31].(15)$$\frac{d}{dt}\left[\begin{array}{c} {\hat{x}}_{2} \end{array}\right]=\left[\begin{array}{c} {\hat{A}}_{21}{\hat{A}}_{22} \end{array}\right]\left[\begin{array}{c} x_{1}\\ {\hat{x}}_{2} \end{array}\right]+\left[\begin{array}{c} {\hat{B}}_{2} \end{array}\right](u+u_{d})$$
where $u_{d}$ is the discontinuous control signal. The goal is for $x_{2}$ to equal ${\hat{x}}_{2}$, so we define the following error signal:(16)$$x_{2e}=x_{2}-{\hat{x}}_{2}.$$
A scalar variable vector was chosen $s(x_{2e})$ (where $s\in R^{n_{2}}$) such that each element of the vector $x_{2e}$ makes $s_{i}{\left(t\right)}^{2}$ a positive definite Lyapunov function, with $s_{i}\left(t\right)=0$ implying $x_{{2e}_{i}}=0$. If $s_{i}\left(t\right)>0$, $s_{i}\left(t\right)$ should decrease, i.e., ${\dot{s}}_{i}\left(t\right)<0$. Conversely, if $s_{i}\left(t\right)<0$, $s_{i}\left(t\right)$ should increase, i.e., ${\dot{s}}_{i}\left(t\right)>0$. Merging these expectations, we obtain the following condition:(17)$${\dot{s}}_{i}\left(t\right)s_{i}\left(t\right)<0.$$
The error signal is affected through $u_{d,i}$. It can be shown that under certain additional conditions, a simple relay intervention can ensure that condition (Equation 17) is met:(18)$$u_{d,i}\left(t\right)=\Gamma_{i}\mathrm{sign}\left(s_{i}\left(t\right)\right),\;\mathrm{where}\;\Gamma_{i}>0.$$
The larger the value of $\Gamma_{i}$, the larger the domain over which the control law for the observer ensures the existence of the sliding mode. Alternatively, (Equation 18) can be interpreted as a PWM signal that has an average equivalent signal $u_{d_{eq}}\left(t\right)$ that maintains $s_{i}\left(t\right)=0$. It can be assumed that $u_{d,i}\left(t\right)$ changes slowly; therefore, it can be approximated by a simple low-pass filter.(19)$$T_{c}^{3}{\hat{u}}_{d_{eq}}^{\left(3\right)}+3T_{c}^{2}{\hat{u}}_{d_{eq}}^{\left(2\right)}+3T_{c}{\hat{u}}_{d_{eq}}^{\left(1\right)}+{\hat{u}}_{d_{eq}}=u_{d,i}\left(t\right),$$
Here, the hat ^^^ on the variable denotes the approximation, ^(*i*)^ indicates the *i*-th derivative with respect to time, and $T_{c}$ characterizes the filter’s cutoff frequency. If this algorithm is implemented with a computer controller, all equations described in this section must be in discrete-time form. The value of $\Gamma_{i}$ can be further reduced by applying the following control law instead of (Equation 18):(20)$$u_{d,i}\left(t\right)={\hat{u}}_{d_{eq},i}\left(t\right)+\Gamma_{i}\mathrm{sign}\left(s_{i}\left(t\right)\right),\;\mathrm{where}\;\Gamma_{i}>0.$$
The details of this are beyond the scope of this paper. Additional information can be found in [29,32], with a mathematically rigorous and theoretical description in [33].

### 3.2. Observer-Based Sliding Mode Control

The primary challenge in sliding mode control is that all engineering models are based on simplifications compared to reality. Certain simplifications can even cause issues in continuous interventions, such as unexpected oscillations, but these are significantly more dangerous in sliding mode control, where the high-frequency component of the discontinuous control signal can resonate with neglected fast dynamics of the system. This can even lead to unstable oscillations. This is qualitatively different from the phenomenon of “chattering” around the sliding surface due to a finite switching frequency.

The essence of the switching strategy is to return the system to the sliding surface immediately upon leaving it. This requires that the originally continuous trajectory has breakpoints. In a system described by differential equations, ${\dot{x}}_{n}\left(t\right)$ must be able to change instantaneously, or more precisely, change sign so that the trajectory can immediately turn back toward the sliding surface.

Consider Figure 2, suppose that, at point $P_{1}$, we detect that the trajectory has left the sliding surface and switched the control signal. However, instead of turning and moving towards the surface along curve *a* (and then “chattering” around the surface), the trajectory continues to diverge along curve *b* due to unmodeled dynamics. This also happens at point $P_{2}$ in the opposite direction, potentially leading to increasing oscillations [34].

This is qualitatively different from the “chattering” phenomenon caused by a finite switching frequency, where the system always turns back toward the sliding surface after switching. In this case, however, it does not, as shown by the difference between curves *a* and *b*.

The essence of the phenomenon is that an unmodeled dynamic appears in the switching strategy. The solution is to ensure that the system switches before the trajectory crosses to the other side of the sliding surface. In this case, the controlled segment must be simplified, but the actual segment is given, so a state observer must be constructed. This is often necessary anyway, as sliding mode control relies on full-state feedback, and not all state variables are usually measurable. In some cases, the state observer can address the “chattering” issues. The state trajectory of the unmodeled dynamic free observer can always be turned back toward the surface at each switch, so the sliding mode controller forces the observer’s state trajectory toward the sliding surface rather than the actual system’s state trajectory. This is shown in Figure 3, where the curve *a* represents the observer’s trajectory.

In this case, the state observer serves as a kind of pulse width modulation unit. The switching surface and control signal are computed from the observed state variables, even if all state variables are measurable [34]. Finally, an observer must be designed so that the actual and observed trajectories remain sufficiently close; then, the actual trajectory moves near to, if not precisely on, the sliding surface, as shown in Figure 3, where curve *b* represents the actual trajectory.

The controller structure is shown in Figure 4. Once the observer structure and parameters are known, the observer–VSS controller loop can establish an ideal sliding mode (VSS = variable structure system).

According to the principle of singular perturbation [35], the fast dynamics of the actual system can be neglected in the observer model. In this way, the number of observed state variables can be significantly reduced. The equations describing the Luenberger observer are as follows:(21)$$\frac{d}{dt}\hat{x}=A_{n}\hat{x}+B_{n}u+O_{2Gn}\left(y-\hat{y}\right),$$(22)$$\hat{y}=C_{n}\hat{x}+D_{n}u,$$
where $\hat{x}$ represents the estimated state; $A_{n}$, $B_{n}$, $C_{n}$, and $D_{n}$ are the nominal system matrices; $O_{2G}$ is the reduced-order state-observer gain matrix; *y* is the output signal; and $\hat{y}$ represents the estimated output, and the index *n* denotes nominal values. Due to the transient effect of neglected fast dynamics, there will be an asymptotically diminishing error between the outputs of the actual system and the observer, but the unmodeled dynamics do not cause oscillation in the movement.

The general rule of thumb is that the value of $O_{G}$ should be selected so that the error in the state observer is diminished with a time constant that is an order of magnitude smaller than the dominant time constant of the system. Therefore, priority is given to the time constant of the estimation error’s decay rather than the exact value of $O_{G}$:(23)$${\dot{x}}_{err}\left(t\right)=\left(A_{n}-O_{2Gn}C_{n}\right)x_{err}\left(t\right).$$
This study’s dual-observer approach successfully handles the nonlinear and time-varying dynamics of the cordless jigsaw system by combining a reduced-order state-space observer with a first-order reference model. Compared to conventional observer-based methods, like the Luenberger observer, which frequently rely on linear approximations and so are unable to effectively manage the complicated behavior of such systems, this approach represents a substantial breakthrough. The dual observer takes into account nonlinear coefficients such as $a_{42}$, $a_{44}$, and $b_{42}$, and allows for accurate state estimation, even in changing dynamic situations.

This approach ensures the stability of the sliding mode control system by reducing instability and compensating for disturbances caused by unmodeled dynamics. The dual nature of the observer, which separates the functions of the low-order observer and the reference model, allows for more accurate observation of the state variables, even during dynamic load changes. This method reduces the “noise” that is a significant problem in sliding mode controllers, especially in high-frequency switching systems, and improves the stability of the system.

Therefore, the use and efficiency of the sliding mode control in compact, portable equipment such as cordless jigsaws can be significantly improved using dual tracking technology. This technology extends the capabilities of the sliding mode to the application area of controllers, especially in nonlinear, energy-efficient control environments, where resource optimization and stability are important.

## 4. Controller Design and Stability Analysis

### 4.1. Application of the Theory in Section 3.1

In this case, Equation (Equation 14) takes the following form:(24)$$\begin{array}{c} \frac{d}{dt}\left[\begin{array}{c} q\\ i\\ \varphi_{m}\\ \omega \end{array}\right]=\left[\begin{array}{cccc} 0 & 1 & 0 & 0\\ 0 & -\frac{R_{a}+R_{b}}{L_{a}} & 0 & -\frac{k_{e}}{L_{a}}\\ 0 & 0 & 0 & 1\\ 0 & a_{42}\left(\varphi_{m}\right) & 0 & a_{44}\left(\varphi_{m}\right) \end{array}\right]\left[\begin{array}{c} q\\ i\\ \varphi_{m}\\ \omega \end{array}\right]+\left[\begin{array}{cc} 0 & 0\\ \frac{1}{L_{a}} & 0\\ 0 & 0\\ 0 & b_{42}\left(\varphi_{m}\right) \end{array}\right]\left[\begin{array}{c} v_{in}\left(t\right)\\ F_{saw} \end{array}\right],\\ a_{42}\left(\varphi_{m}\right)=\frac{k_{m}}{J_{m}+J_{k}k_{mk}^{2}+m_{a}e_{k}^{2}k_{mk}^{2}\sin^{2}\left(k_{mk}\varphi_{m}\right)},\\ a_{44}\left(\varphi_{m}\right)=-\frac{b+0.5(m_{a}e_{k}^{2}k_{mk}^{3}\sin\left(2k_{mk}\left(\varphi_{m}\right)\right)\omega }{J_{m}+J_{k}k_{mk}^{2}+m_{a}e_{k}^{2}k_{mk}^{2}\sin^{2}\left(k_{mk}\varphi_{m}\right)},\\ b_{42}\left(\varphi_{m}\right)=\frac{-k_{mk}e_{k}\sin\left(k_{m_{k}}\varphi_{m}\right)}{J_{m}+J_{k}k_{mk}^{2}+m_{a}e_{k}^{2}k_{mk}^{2}\sin^{2}\left(k_{mk}\varphi_{m}\right)}. \end{array}$$
In this model, $v_{in}\left(t\right)$ represents the motor input voltage, which is used instead of the constant terminal voltage $U_{0}$, and $F_{saw}$ denotes the measured sawing force. The key perturbed state variable is identified as ω. Consequently, as per Equation (Equation 15), a reduced-order disturbance observer must be constructed. This is achieved by neglecting the electrical time constant $T_{e}$ and focusing the reduced-order observer on the mechanical subsystem, specifically the mechanical submodel.

By definition, the electrical and mechanical time constants of a DC drive are derived as follows, based on the parameters illustrated in Figure 1:(25)$$T_{e}=\frac{L_{a}}{\left(R_{a}+R_{b}\right)},$$(26)$$T_{m}=\frac{\left(R_{a}+R_{b}\right)J_{eq}}{k_{e}k_{m}},$$
where $J_{eq}=J_{m}+J_{k}k_{mk}^{2}$. These time constants also appear in the transfer function relating the angular velocity to the input voltage of the motor:(27)$$\frac{\omega (s)}{v_{in}\left(s\right)}=\frac{\frac{k_{m}}{J_{eq}L_{a}}}{s^{2}+\frac{R_{a}+R_{b}}{L_{a}}s+\frac{k_{e}k_{m}}{L_{a}J_{eq}}}=\frac{\frac{1}{k_{e}}}{\frac{L_{a}J_{eq}}{k_{e}k_{m}}s^{2}+\frac{\left(R_{a}+R_{b}\right)J_{eq}}{k_{e}k_{m}}s+1}.$$
Nominal motor parameters indicate that $T_{m}$ is approximately an order of magnitude larger than $T_{e}$, as shown in Table 1. While these expressions for the time constants differ slightly from the actual values, the discrepancy is negligible if $T_{m}\gg T_{e}$. The first-order reference model equation can be expressed and referred to as a reduced-order state observer:(28)$$\frac{d\omega (t)}{dt}=-\frac{\omega (t)}{T_{m}}+\frac{1}{k_{e}}v_{in}\left(t\right).$$
The requirement was to slow down the motor transients and thus reduce the current peaks during transients. This was achieved by virtually increasing the mechanical time constant of the motor, in other words, by forcing the motor to follow the virtual model described by the following equation using sliding mode control. Based on Equation (Equation 14), the parameters of the system model were changed only for the state variable that the input signal directly affects. In the study, this was the speed; i.e., a virtual model, in other words an observer, for the speed with the changed time constant according to (Equation 15), so we will refer to this model later as the first-order observer.(29)$$\frac{d\hat{\omega }\left(t\right)}{dt}=-\frac{\hat{\omega }\left(t\right)}{{\hat{T}}_{m}}+\frac{1}{k_{e}}v_{in}\left(t\right).$$
For the Simulink model, the equation was written in transfer function form (remark: here, the *s* is the Laplace operator):(30)$$\frac{\hat{\omega }\left(s\right)}{v_{in}\left(s\right)}=\frac{1/k_{e}}{{\hat{T}}_{m}s+1},$$
where ${\hat{T}}_{m}=factorT_{m}$ (remark: according to (Equation 26), this means ${\hat{J}}_{m}=factorJ_{eq}$). The scalar variable representing the velocity modeling error is defined as(31)$$s\left(t\right)=\omega \left(t\right)-\hat{\omega }\left(t\right).$$
Since the drive operates in a single quadrant, a negative voltage cannot be applied to the motor. Hence, Equation (Equation 20) is revised as follows:(32)$$u_{d}\left(t\right)={\hat{u}}_{d_{eq}}+\frac{1}{2}\Gamma +\frac{1}{2}\Gamma \mathrm{sign}\left(s\left(x\right)\right),\;\mathrm{where}\;\Gamma >0.$$
Here, ${\hat{u}}_{d_{eq}}$ is generated by a filter defined as Equation (Equation 19). In sliding mode, where s(t)=0, it follows that $\dot{s}\left(t\right)=0$. The equivalent control signal is estimated under the condition $\dot{s}\left(t\right)=0$ by evaluating the parameter difference in the differential equations of $\dot{\omega }$ and $\dot{\hat{\omega }}$. Additional external disturbances d(t) can be considered if they meet the criteria detailed in engineering approaches [29] or mathematical approaches [33]. In this enhanced methodology, the theoretical model is directly linked to practical challenges such as minimizing current spikes, maintaining stability under varying load conditions, and optimizing energy efficiency. The sliding mode control framework combined with disturbance observation ensured robust and efficient motor operation, aligning the simulation results with the theoretical predictions.

### 4.2. Condition for Sliding Mode Stability

To rigorously prove the stability condition, we define a Lyapunov function for the sliding surface s(t):(33)$$V\left(s\right)=\frac{1}{2}s^{2}\left(t\right),$$
where V(s) is positive definite, since V(s)>0 for all s(t)≠0, and V(s)=0 if s(t)=0. This satisfies the requirement for a Lyapunov function. The time derivative of the Lyapunov function is(34)$$\dot{V}\left(s\right)=s\left(t\right)\dot{s}\left(t\right).$$
The negative definiteness of the Lyapunov function derivative ($\dot{V}\left(s\right)<0$) ensures that the system’s trajectories are attracted to the sliding surface s(t)=0, thereby guaranteeing stability under all conditions. This condition is equivalent to(35)$$s\left(t\right)\dot{s}\left(t\right)<0,$$
which ensures that the system trajectories converge to the sliding surface s(t)=0. This ensures that the system always returns to the sliding surface and remains there. If s(t)>0, then $\dot{s}\left(t\right)<0$, and vice versa. Below, it is shown how this condition can be ensured for the jigsaw model.

In this case, the sliding surface is defined as the difference between the angular velocity and the reference angular velocity according to (Equation 31). The $\dot{s}\left(t\right)$ must be estimated by(36)$$\dot{s}\left(t\right)=\dot{\omega }\left(t\right)-\dot{\hat{\omega }}\left(t\right).$$
According to (Equation 29), the equation of $\dot{\hat{\omega }}\left(t\right)$ is known. The system dynamics can be described by the following equations, taking into account the properties of the motor and electrical components. In Equation (Equation 24), the rank of the 4×4 system matrix is only 2. It means that the state variables *q* and $\varphi_{m}$ can be omitted. Since $T_{e}$ << $T_{m}$, the state variable *i* is also eliminated by the selection $L_{a}=0$; then, the current *i* can be expressed as follows:(37)$$i=\frac{1}{R_{a}+R_{b}}\left(v_{in}+u_{d,eq}\right)-\frac{k_{e}\omega }{R_{a}+R_{b}}.$$
The the nonlinear coefficients are split into a constant part and an incremental part, and the increments are expressed as follows:(38)$$∆a_{42}=a_{42}\left(\varphi_{m}\right)-a_{42},$$(39)$$∆a_{44}=a_{44}\left(\varphi_{m}\right)-a_{44},$$
The fourth scalar differential equation in the matrix Equation (Equation 24), considering (Equation 38) and (Equation 39), takes the following form:(40)$$\dot{\omega }=\left(a_{42}+∆a_{42}\right)i+\left(a_{44}+∆a_{44}\right)\omega +0v_{in}+b_{42}F_{saw}.$$
Substituting Equations (Equation 29), (Equation 37) and (Equation 38) into Equation (Equation 36):(41)$$\dot{s}=\dot{\omega }-\dot{\hat{\omega }}=∆a_{42}\left(\frac{v_{in}\left(t\right)+u_{d,eq}}{R_{a}+R_{b}}-\frac{k_{e}\omega }{R_{a}+R_{b}}\right)+∆a_{44}\omega +b_{42}F_{saw},$$(42)$$\dot{s}=\frac{∆a_{42}}{R_{a}+R_{b}}\left(v_{in}\left(t\right)+u_{d,eq}\right)+\left(-\frac{∆a_{42}k_{e}}{R_{a}+R_{b}}+∆a_{44}\right)\omega +b_{42}F_{saw}.$$
Equation (Equation 32) defines the switching strategy of the control system based on the positive or negative values of the sliding surface *s*. When s>0, the control action increases by Γ, while for s<0, it remains at the equivalent $u_{d_{eq}=0}$ value.

According to Equations (Equation 32) and (Equation 35), the following inequalities can be derived:(43)$$s\left(t\right)\left[\frac{∆a_{42}}{R_{a}+R_{b}}\left(v_{in}\left(t\right)-\frac{1}{2}\Gamma -\frac{1}{2}\Gamma sign\left(s\right)\right)+\left(-\frac{∆a_{42}k_{e}}{R_{a}+R_{b}}+∆a_{44}\right)\omega +b_{42}F_{saw}\right]<0.$$
By giving an upper estimate for the state variables and the unknown parameters, a condition can be defined for the value of Γ:(44)$$\Gamma >-v_{in}\left(t\right)+k_{e}\omega -\frac{∆a_{44}\left(R_{a}+R_{b}\right)}{∆a_{42}}\omega -\frac{\left(R_{a}+R_{b}\right)}{∆a_{42}}b_{42}F_{saw}.$$
The calculations for Γ with 0 −$\frac{\omega_{max}}{2}$ and $k_{mk}\varphi_{m}=\frac{\pi }{2}$ the $F_{saw}$, and the following results:(45)$$\Gamma >-18+7.383\cdot 10^{-3}\cdot \frac{2445}{2}-\frac{0.8187\cdot (0.176+0)}{228.5}\cdot \frac{2445}{2}-\frac{\left(0.176+0\right)}{228.5}\cdot \left(-43\right)\cdot 400,$$(46)Γ>−18+9.030735−0.7858176+13.257036,(47)Γ>3.503.
The calculations for Γ with $\omega_{max}$ and $k_{mk}\varphi_{m}=\frac{\pi }{2}$ the $F_{saw}$, and the following results:(48)$$\Gamma >-18+7.383\cdot 10^{-3}\cdot 2445-\frac{0.8187\cdot (0.176+0)}{228.5}\cdot 2445-\frac{\left(0.176+0\right)}{228.5}\cdot \left(-43\right)\cdot 400,$$(49)Γ>−18+18.049935−1.5716352+13.278468,(50)Γ>11.758.
It can be seen that the higher the rotational speed, the higher the Γ. By satisfying Γ>11.78V, the sliding mode controller compensates for variations in $a_{42}$, $a_{44}$ and $b_{42}$, as well as external disturbances ($F_{\mathrm{saw}}$), ensuring that the Lyapunov stability condition ($\dot{V}\left(s\right)<0$) holds. This analysis shows that the dynamic variations in these parameters, along with the nonlinear and time-varying nature of the jigsaw’s electromechanical system, are explicitly handled, thereby ensuring robust stability under all operating conditions.

This analysis shows that the dynamic variations in $a_{42}$, $a_{44}$ and $b_{42}$, along with external disturbances ($F_{\mathrm{saw}}$), are explicitly handled by the derived sliding mode controller, ensuring robust stability, while fulfilling the Lyapunov stability condition under nonlinear and time-varying conditions.

## 5. Experimental System Description

### 5.1. Presentation of the Measurement Arrangement

The purpose of the measurement was to examine and optimize the jigsaw’s machining process in detail, focusing specifically on control engineering regulation options. As a first step, in the measurement, the cutting force was determined based on the jigsaw’s load under various machining conditions. These force measurements provided a detailed understanding of the tool’s power consumption and required energy levels.

The cordless jigsaw was then subjected to a control engineering regulation task employing sliding mode control, which was dependent on the measured data. With modifications continuously adjusting to the actual cutting circumstances, this control system allowed the cordless jigsaw to achieve the optimal current draw and steady operation. The control’s objective was to ensure that the cordless jigsaw operated as efficiently and with the least amount of vibration possible, stabilizing the tool’s performance, even while cutting materials with different hardness levels.

The information gathered in this manner was arranged in tables, and the behavior of the system was examined to improve the jigsaw’s effectiveness even further. The outcomes of the measurement assessment make it evident how well the control worked and how the system reacted to machining factors. In order to assess the jigsaw’s functioning in a simulated model, the evaluated data were used to construct a sliding mode controller using MATLAB and Simulink. This method allowed the numerous control settings to be fine-tuned before being implemented on the actual tool, in addition to optimizing the system’s stability and current consumption.

### 5.2. Hardware Implementation

Figure 5 shows the hardware implementation of the measurement arrangement.

The test bench operated in this manner: The test specimen to be sliced was put onto a specially made carriage system that could only move vertically. Rolling bearings, which lower friction and guarantee smooth carriage movement, assisted this system. Because it guaranteed that the carriage can only move vertically, which is the direction in which the cutting force works, this design was crucial. A load cell, which continually measured the cutting force and output a signal proportionate to it, also supported the carriage. Because it made precise force monitoring throughout the cutting process possible, this load cell was an essential part of the test bench.

Because undesired vibrations have the potential to skew readings and lower system accuracy, special emphasis was given to limiting them throughout the test bench’s design. To achieve this, a lead screw was preloaded into the carriage system, creating a constant force, $F_{c}$, between the carriage and the load cell when it is at rest. By lowering unwanted vibrations, this preload force helped maintain the carriage in its starting position and improved the measurement accuracy.

A test specimen of 80×80×80mm wood was fastened into the carriage system prior to the cutting trials starting. By keeping the test piece in line with the grain of the wood, cuts were produced perpendicular to the grain structure. This positioning allowed for a smooth cut, as the saw blade cut against the grain of the wood, and provided relevant information about the amount of cutting force. The planned movement direction of the carriage is indicated by a dark arrow in Figure 5, which also shows the direction of the cutting force measured during the operation.

The cordless jigsaw (designated as (6)) was switched on during the experiment and an inductive sensor (designated as (1)) monitored its oscillation mechanism. This sensor continuously logged the saw speed in the measuring system. Due to this, the sensor provided accurate information about the saw blade movement speed.

A three-phase motor (designated as (4)) provided the feed movement, which was operated by a frequency converter and allowed the speed to be varied according to the cutting process. The feed speed of the jigsaw could be finely adjusted thanks to the variable speed. An extra inductive sensor (designated as (3)) was placed to measure the feed motor’s rotation speed. It sensed the uniformly spaced 6 mm diameter screws positioned on the feed motor’s pulley when it was rotating. Accurate management of the feed rate during cutting was made possible by the exact measurement of the motor’s speed using these evenly spaced screws.

Figure 6 illustrates the course of the cutting force and the angular velocity as a function of time, which determined the cutting speed.

The red curve represents the measured cutting force, considered an external disturbance. This force acted on the tool during the cutting process, influencing both the cutting quality and the motor’s current draw.

The blue curve shows the saw blade’s speed during cutting, with a peak value of 2.72 m/s. Since the speed curve was on a different scale than the cutting force, the values displayed for speed were obtained by multiplying the shown scale by 0.0054. This scaling allowed both functions to appear on the same graph, illustrating the relationship between the cutting force and speed across the different stages of the cutting process.

The cutting force and saw blade speed both exhibited time-varying oscillations at different stages of the cutting process, as can be seen by analyzing the curves. This oscillating activity resulted from variations in the resistance that the blade encountered during cutting, as it required different amounts of force to enter and depart the material. The peak speed and cutting force were correlated, which offered important information about the energy efficiency of the cutting process and the dynamic properties of the system.

In addition to showing how saw blade speed and cutting force interacted, this image aids in locating the locations of greatest load throughout the cutting process. This data make it possible to optimize the cutting process, since a better speed and force profile would results in a more economical energy consumption and a longer tool life.

Figure 7 illustrates the trajectories of the cutting force as a function of the saw blade speed.

The figure illustrates the course of each cycle using curves that create closed loops when moving counterclockwise.

Each closed loop corresponds to a single cutting cycle, representing the forward and backward movement of the saw blade as it penetrated and exited the material. Observing the trajectory, we can see that cutting force changed with the speed at different stages of the cycle. During the forward motion, as the blade penetrated deeper into the material, the cutting force typically increased, then decreased as the blade exited the cut.

This trajectory provides a detailed view of the dynamic changes occurring during cutting and helps in understanding how the cutting resistance influences the blade’s speed and the forces acting on the tool. Such a representation is valuable for optimizing the cutting process, as analyzing force–speed trajectories offers insights into how the load varies across different stages of the cycle, impacting the energy consumption and tool lifespan.

## 6. Simulation Setup

The simulation program developed under the Simulink system contained four models running in parallel. Figure 8 shows the top block, which contained a two-state, linear reference model that ran uncompensated and independently of the other models.

In the block diagram, the second block non-linearly models the cutting process, which was uncompensated and ran independently of the other models. The third and fourth blocks did not run independently of each other. The third block’s content was identical to the second, so it also described a non-linear cutting process. This block was compensated through the fourth simplified first-order reference model.

The contents of the four different blocks are shown in detail below. The second-order reference model is illustrated in Figure 9.

An uncompensated non-linear cutting model block diagram is shown in detail in Figure 10.

The model illustrates the internal structure of the system, where the input signals (voltage *v* and force *F*) influenced the calculation of the output variables (position *q*, current *i*, angular displacement φ, and angular velocity ω).

This block determines the system’s responses based on the state variables and input signals. The model includes multiple feedback loops that calculate the parameters $a_{42}$, $a_{44}$, and $b_{42}$ based on the output variables. These parameters function as non-linear terms, capturing additional dynamics of the system. The f(u) blocks compute these non-linear parameters, taking into account the system’s dynamic characteristics. For example, the parameters $a_{42}$ and $a_{44}$ are influenced by the system’s mechanical and electrical properties, while the $b_{42}$ block is based on feedback from the cutting force.

The inputs and outputs are denoted symbolically, with an input matrix representing the inputs and output blocks representing the outputs. Each feedback path and parameter plays a crucial role in defining the dynamic behavior, especially in a non-linear system like the cordless jigsaw model. The compensated non-linear model is identical to the uncompensated non-linear model (see Figure 10).

The blocks shown in Figure 11 provide a detailed view of the contents of the $a_{42}$, $a_{44}$, and $b_{42}$ elements. These blocks offer a more in-depth insight into the operation of the model, especially in the reduced-order disturbance detection process. The $a_{42}$ and $a_{44}$ blocks contribute to modeling the motor state variables and examining the stability conditions of the sliding mode control, while the $b_{42}$ block represents the external forces acting on the saw, which influence the system’s dynamics. The simplified first-order reference model is illustrated in Figure 12.

The first-order reference model estimates the motor’s angular velocity $\hat{\omega }$ using a modified transfer function, where $k_{e}$ represents the motor constant, $T_{m}$ is the mechanical time constant, and the parameter “factor” scales the time constant. This setup allows the observer to respond faster or slower, depending on the control requirements. By adjusting the responsiveness, the system better aligns the observed states with the actual motor dynamics, which is particularly useful when different levels of responsiveness are required. The parameters used in the simulation for the electromechanical model of the cordless jigsaw are summarized in Table 1.

### 6.1. Sliding Mode Simulation

The simulation program modeled the operation of the jigsaw for 2 s. The differential equations were integrated with a time step of $10^{-6}$ s using an Euler solver. The results for the three types of models—the reference, uncompensated non-linear model, and compensated non-linear model—are illustrated in the common diagrams in red, blue, and black, respectively.

The angular velocities are illustrated in Figure 13. The uncompensated non-linear model reached operating speed in approximately 0.25 s. The process modeled with sliding compensation took significantly longer to reach operating speed. Thanks to the sliding model, the compensated model achieved an angular velocity function close to the reference model. The slow current ramp-up allowed a significantly lower current to flow in the jigsaw at startup.

Figure 13 compares the angular velocity of the three models: reference (red), uncompensated non-linear (blue), and compensated non-linear (black). The compensated model demonstrated a smoother convergence to the reference angular velocity, while reducing the current spikes at startup.

The slower speed ramp-up in the compensated model was due to the sliding mode control, which ensured a gradual current draw, reducing the stress on the components and battery during startup. The close alignment with the reference model demonstrated the effectiveness of the sliding mode control and the dual observer in stabilizing the system under dynamic conditions. Reducing current spikes at startup is critical for improving the energy efficiency and lifespan of battery-powered tools.

This result highlights the advantages of sliding mode control and the dual-observer system in achieving system stabilization and energy efficiency. The figure clearly illustrates how the proposed methodology improved the system behavior during the startup phase, aligning with the research goals of optimizing the cordless jigsaw’s performance.

Figure 14 shows the current draws for the three models, which produced completely different functions. Due to the sliding model, the peak current of the compensated model was approximately half that of the uncompensated non-linear model. The purpose of this approach was to achieve a lower current peak at startup, accepting a slower rise in speed as a compromise. This method extended the lifespan of the cordless jigsaw’s electric motor by reducing the risk of overheating. Additionally, the calculated current draw remained unstable during the simulation run because of the different dynamics of the system model and first-order reference model. In other words, the unmodeled dynamics in the system model, from the point of view of the first-order reference model, caused the measured diagram to exhibit persistent oscillations rather than diminishing ones.

Figure 14 illustrates the current–time diagram for the reference (red), uncompensated (blue), and compensated (black) models. The uncompensated non-linear model exhibited significant current spikes, reaching approximately 80 A during startup. In contrast, the compensated model reduced the peak current to around 40 A, demonstrating the effectiveness of the sliding mode control in mitigating excessive current draw. This reduction not only minimized the thermal stress on the motor but also enhanced the overall efficiency and safety of the system.

The oscillatory behavior observed in the compensated model was a result of the interaction between the sliding mode controller and the unmodeled dynamics in the system. While the controller successfully limited the peak current, the mismatch between the system model and the reference model introduced oscillations that persisted throughout the simulation. These oscillations highlighted the need for further refinement of the controller to better account for the system dynamics.

This result aligns with the primary objective of the research, which aimed to balance the trade-off between current peak reduction and system stability. The sliding mode control framework, combined with a dual-observer structure, demonstrated its potential in addressing these challenges effectively, while improving the lifespan and operational efficiency of cordless jigsaws.

Figure 15 illustrates the relay switching that enables sliding mode operation. Initially, switching occurs at a low frequency. After one second, the switching frequency reaches 3 kHz. In practice, this switching frequency can be achieved without issues in power-regulation electronics.

When using a first-order reference model in a simulation, the unmodeled dynamics present significant challenges that can lead to instability, causing the simulation to “blow up”. Unmodeled dynamics refer to those components and effects that the mathematical model cannot fully capture or account for. These elements, which may include nonlinear or higher-frequency dynamics, though small in their individual impact, can accumulate over time, resulting in growing discrepancies between the observed and actual system behavior.

In sliding mode control, the first-order reference model estimates the system variables (such as the motor’s angular velocity and current draw) affected by unmodeled dynamics. As these discrepancies accumulate, the estimated values increasingly deviate from the true states, and the model looses coherence with the real system behavior, especially regarding the current–time and angular velocity–time functions.

While sliding mode control effectively reduces peak currents at startup and prolongs motor lifespan, instability due to unmodeled dynamics becomes evident during observations. As the unmodeled dynamics begin to dominate, an observer-based simulation may produce oscillations or abrupt fluctuations, instead of the expected stable current or angular velocity profile. Ultimately, this can lead to the simulation’s collapse, as the deviations grow exponentially.

### 6.2. Sliding Mode Simulation with Additional Reduced-Order State-Space Observer

Figure 16 shows the setup of the simulation model with the integration of a reduced-order state-space observer. By introducing this observer, the system dynamics benefited from a dual-layered observation approach, which enhanced the stability and optimized the performance.

The reduced-order state-space observer works in conjunction with the first-order reference model, providing a more accurate estimation of the system states, especially in the presence of dynamics that are not fully captured by the model (unmodeled dynamics). This setup minimizes deviations between the observed and actual system behaviors, reducing the oscillations that often arise from high-frequency switching in sliding mode control. The reduced-order state-space observer thus ensures more accurate tracking of the reference values and mitigates the effects of unmodeled dynamics, resulting in a smoother and more stable system operation. The time constant of error elimination can be calculated based on (Equation 23):(51)$$\dot{\omega }-\dot{\hat{\omega }}=\left(-\frac{1}{T_{m}}-O_{2G}\right)\left(\omega -\hat{\omega }\right).$$
The time constant of error elimination:(52)$$T_{err}=-\frac{1}{-\frac{1}{T_{m}}-O_{2G}}=0.03.$$
Figure 17 shows the content of the reduced-order state-space observer.

This observer directly estimates the motor’s angular velocity without additional modifications to the time constant, providing a baseline for the estimation. Together with the first-order reference model, this configuration enables dual-layer state estimation, which enhances the stability and accuracy by compensating for dynamic variations, ultimately increasing the overall robustness and control reliability of the system.

In Figure 18, the angular velocity–time diagram compares three different control cases: the reference model (red), the uncompensated non-linear model (blue), and the compensated model with a reduced-order state-space observer (black). The addition of the reduced-order state-space observer significantly improved the angular velocity response, bringing the compensated model closer to the reference.

The results demonstrate that the reduced-order state-space observer enhanced the system performance by stabilizing the angular velocity fluctuations and ensuring a smoother transition to the reference value. In contrast to the unbalanced model, which quickly reached the operating speed but exhibited large deviations and oscillations, the balanced model provided a more controlled and gradual convergence to the desired angular velocity. This change improved the reliability of the tool and reduced the mechanical stress.

The low-order observer reduced vibrations, which are the main problem in sliding mode control systems. The smoother transitions result in lower power consumption and wear of mechanical components, so this method not only improves the dynamic stability but also saves energy. These results suggest the incorporation of reduced-sequence state-space observation into sliding mode control systems for nonlinear dynamic systems, such as cordless jigsaws. The observer highlights the useful advantages of the proposed method in improving the efficiency and stability of the system, as well as providing flexibility and accuracy in observing reference values by compensating for unmodeled dynamics. The current–time diagram in Figure 19 illustrates the impact of the reduced-order state-space observer on the current stability.

During operation, unbalanced systems exhibit significant oscillations, with current peaks of up to approximately 80 A. These oscillations indicate instability, which leads to increased component wear and reduced operating efficiency. When low-order state-space observers were considered, the system exhibited a significantly smoother current response.

With the compensation model, current stabilization was achieved in a more controlled and efficient manner. Peak currents were reduced to almost half of the uncompensated value. This stabilization is essential for extending the service life of cordless jigsaws. The stabilization system reduces current peaks and improves the supervision of the reference current. The system improves reliability and efficiency by reducing the energy losses during the short start-up phase. The results clearly demonstrate the important role of low-order state-space supervision in maximizing the efficiency of sliding mode drive systems. They successfully address issues such as vibration and unmodeled dynamics and guarantee the stability and efficiency of battery-powered equipment.

Figure 20 shows the angular velocities of the cordless jigsaw model with the red line and the reduced-order state observer with the blue line. The close fit of the two curves shows how well the low-order state observer followed the dynamics of the system. By correcting for the unmodeled dynamics, the observer was able to ensure that the assumed angular velocity was approximately the same as the true angular velocity of the puzzle model. This correction shows that, despite perturbations and modeling errors, the low-order observer represented the fundamental behavior of the system well; the small difference between the two curves suggests that the observer did not reproduce the system behavior accurately. The deviations were due to the unmodeled high-frequency dynamics and disturbances. Despite the small deviations, the observer made robust and reliable predictions, improving the stability and performance of the system. Observers provide accurate state feedback by reducing the deviation between the predicted and actual outputs. These results emphasize the benefits of incorporating low-order observers into control systems to achieve accuracy and reliability in dynamic operation. As a result of the Scotch yoke mechanism, the lower order observer’s curve was comparatively smooth, while the jigsaw model’s curve oscillated around it. The zoomed-in part of Figure 20 is displayed in Figure 21.

The vibrations of the nonlinear model of the jigsaw were filtered out by the reduced-state observer. Therefore, the sliding mode control did not cause high-frequency, unmodeled dynamics. Figure 21 shows how the reduced state observer behaved differently from the output of the cordless jigsaw model.

The dynamics created by the Scotch yoke mechanism caused repetitive oscillations in the system output, which were responsible for the oscillatory nature of the jigsaw model. These oscillations were effectively smoothed by the reduced serial-state observer, providing a stable and reliable reference for the sliding mode control. This smoothing effect is necessary to prevent the excitation of unmodeled high-frequency dynamics, which can cause system instability and poor performance. The reduced serial-state tracking filters out these oscillations, eliminating the unwanted high-frequency deviations of the sliding mode control, reducing wear on system components and improving energy efficiency. In addition, the output of the filtered observer more closely matches the desired reference signal, providing accurate and reliable control.

## 7. Conclusions

This paper presented the design and use of a sliding mode control system designed to address the operational challenges of cordless jigsaws, such as minimizing peak current spikes, extending motor life, and improving energy efficiency. The proposed system handles the complex nonlinear dynamics of the system and provides stable, reliable, and efficient operation over a wide range of load conditions thanks to a novel dual-observer design.

The dual-observer design, which combines a first-order reference model and a low-order state-space observer, offers significant advantages in stabilizing the system and compensating for unmodeled dynamics. The first-order reference model simplifies the control procedure and eliminates high-frequency, unmodeled dynamics that cause instability and resonance. This multi-layered monitoring strategy reduces the risk of overheating and mechanical wear by minimizing current surges, optimizing power consumption, and accurately tracking the angular velocity.

The results showed how the dual-observer control architecture reduced the typical error of sliding mode control, the “chattering” effect. By reducing the negative effects of high-frequency switching, the mechanical lifetime of the system was improved, while maintaining control accuracy. The proposed technology proved its robustness in practical applications by operating continuously and efficiently under dynamically changing load conditions. Although the initial run time was slightly longer due to the initial slow speed increase, the mechanical stress and wear were significantly reduced, thus guaranteeing the long lifetime and reliability of the system.

Despite the advantages of dual-observer structures, it must be acknowledged that they have limitations. Low-order state observers may have difficulty compensating for high-frequency, unmodeled dynamics, especially in the presence of strong disturbances or severe operating conditions. Furthermore, the accuracy of the system model may affect the observer performance, and parameter changes or faulty modeling may affect the observer operation. These shortcomings highlight the need for further research in adaptive observer-based tactics that dynamically respond to changing system properties and external perturbations, increasing the flexibility and utility of the proposed method.

The incorporation of a dual-observer structure, the reduction in peak current peaks, and the development of new sliding mode stability criteria are the main contributions of this work. These developments address the challenges of unmodeled dynamics and provide reliable and energy-efficient solutions for battery-powered devices.

In conclusion, the proposed dual-monitoring structure represents a significant step forward in achieving the best possible performance for sliding mode systems, especially when unmodeled dynamics are present. This approach guarantees energy efficiency and improves the stability, accuracy, and reliability of the system, while maintaining mechanical integrity. These results provide a solid foundation for future development of battery-powered, energy-efficient instruments.

Future work should focus on investigating multilayer monitoring systems and developing the proposed methods for applications requiring higher performance and more complex dynamics. Furthermore, adaptive monitoring techniques could be used to increase the flexibility and enable efficient operation.

## Figures and Tables

**Figure 1 sensors-25-00456-f001:**
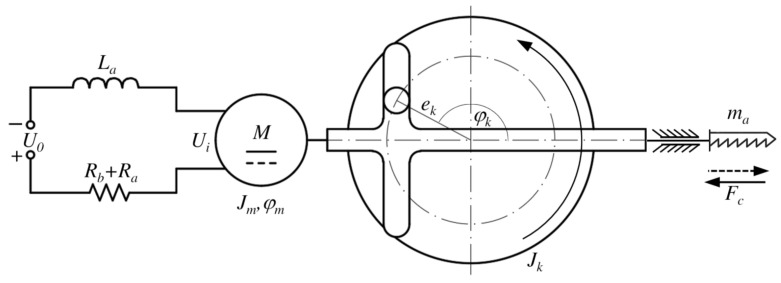
Operating principle of a jigsaw mechanism.

**Figure 2 sensors-25-00456-f002:**
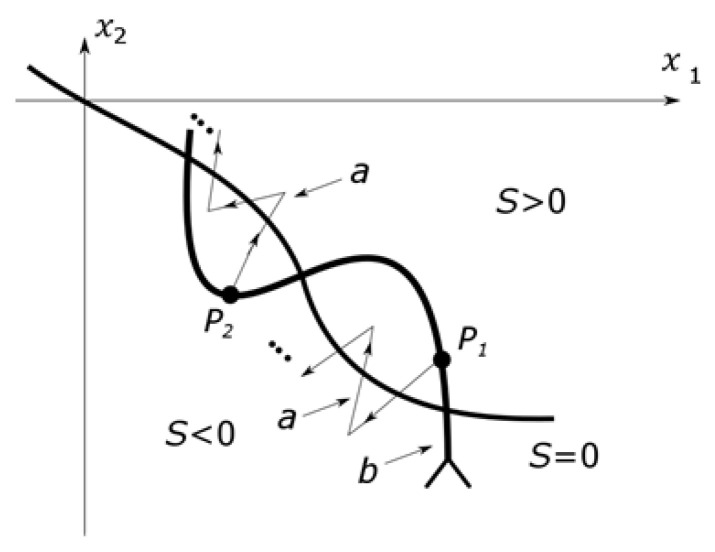
The effect of unmodeled dynamics on the connection.

**Figure 3 sensors-25-00456-f003:**
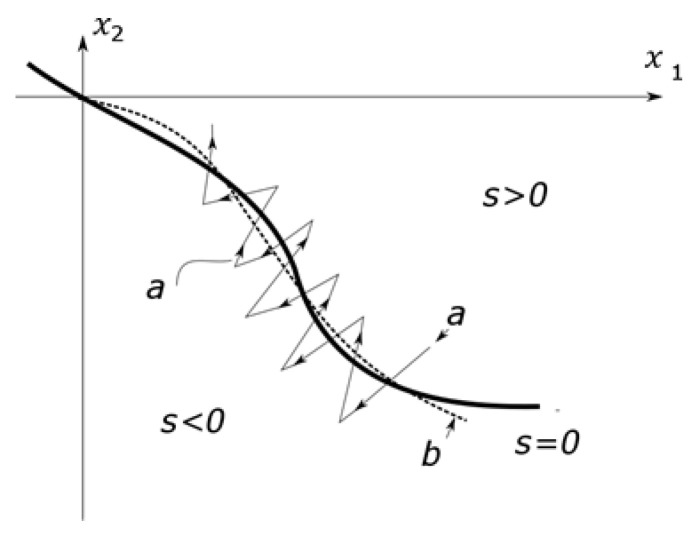
State trajectory of a sliding mode controller based on an unmodeled free observer.

**Figure 4 sensors-25-00456-f004:**
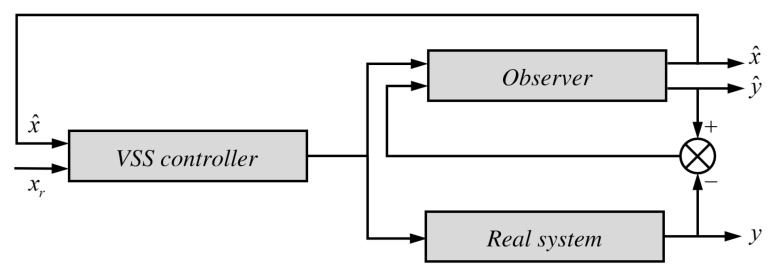
Control structure of a variable structure system.

**Figure 5 sensors-25-00456-f005:**
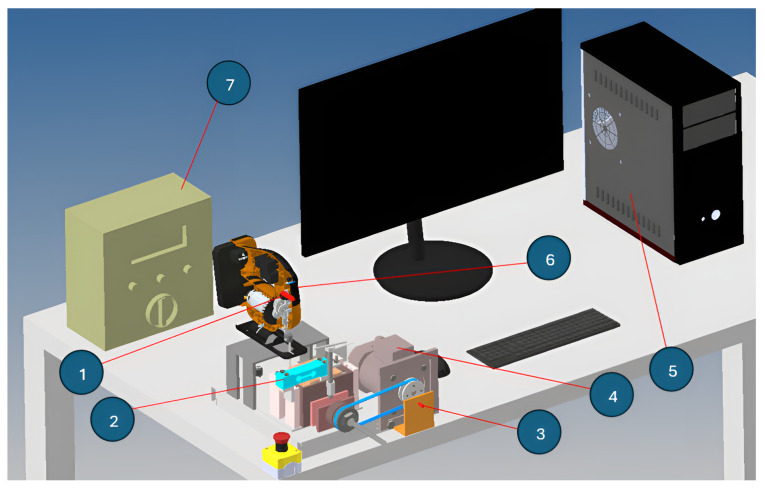
Measuring arrangement (1—inductive transmitter, 2—load cell, 3—inductive transmitter, 4—advance motor, 5—PC, 6—cordless jigsaw, 7—data collection unit).

**Figure 6 sensors-25-00456-f006:**
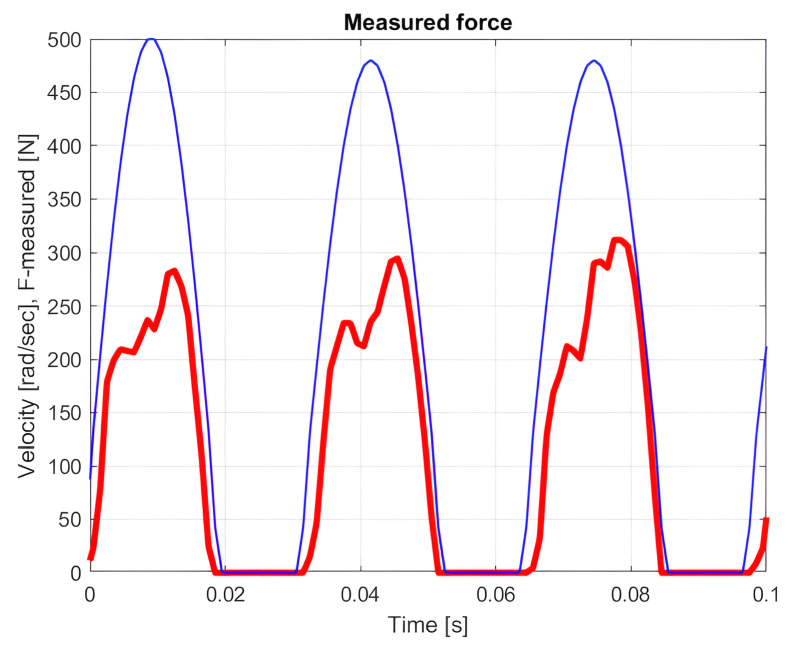
The cutting force–time function, with the speed–time function’s scale being 0.0054 times that of the force scale.

**Figure 7 sensors-25-00456-f007:**
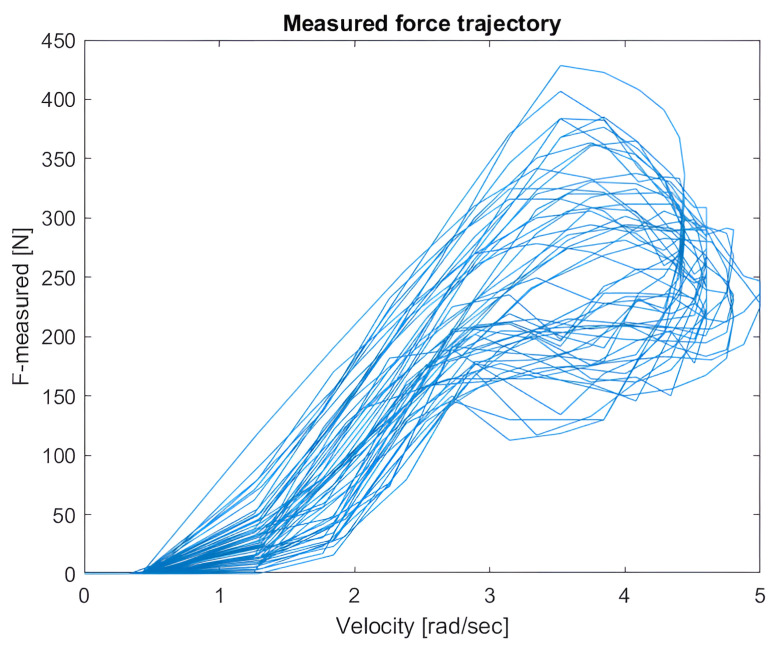
Cutting force trajectory.

**Figure 8 sensors-25-00456-f008:**
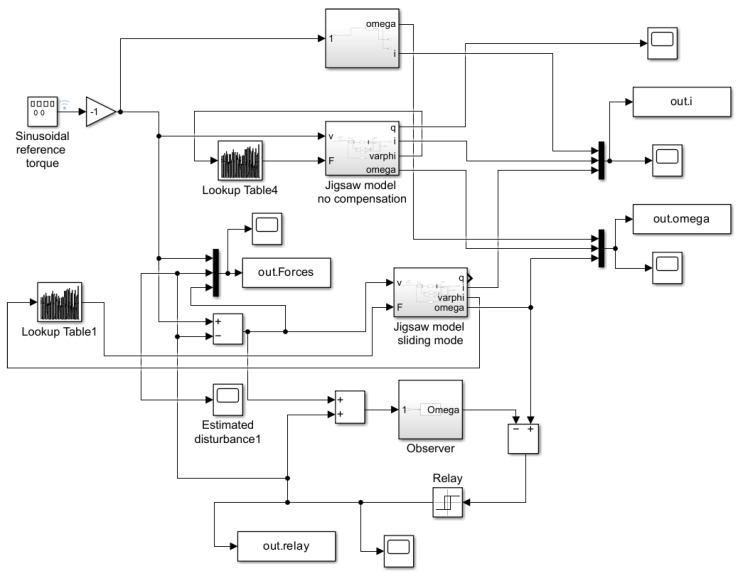
System block diagram.

**Figure 9 sensors-25-00456-f009:**
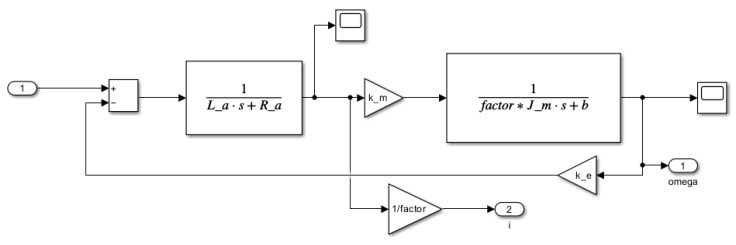
Second-order reference model according to (Equation 27) and ${\hat{J}}_{m}=factorJ_{eq}$ (in the other words: ${\hat{T}}_{m}=factorT_{m}$) selection.

**Figure 10 sensors-25-00456-f010:**
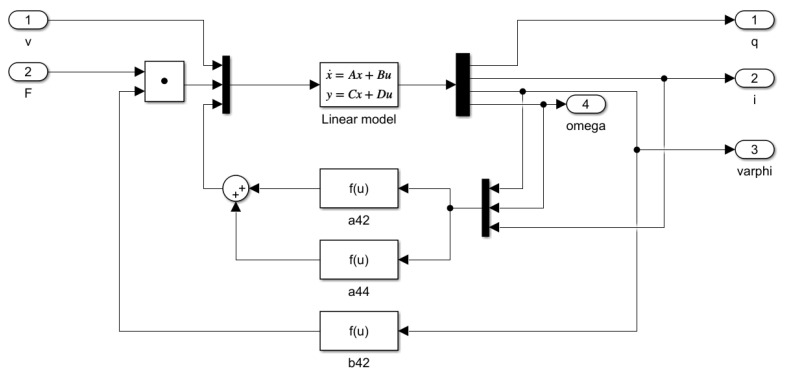
Uncompensated non-linear model.

**Figure 11 sensors-25-00456-f011:**
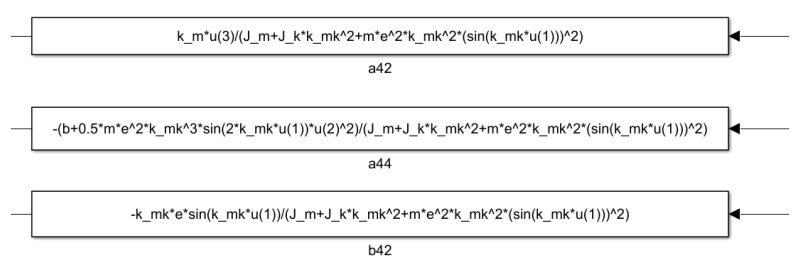
The contents of blocks $a_{42}$, $a_{44}$ and $b_{42}$ are shown in detailed form according to (Equation 24).

**Figure 12 sensors-25-00456-f012:**
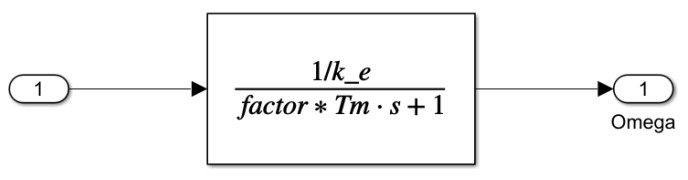
First-order reference model according to (Equation 30).

**Figure 13 sensors-25-00456-f013:**
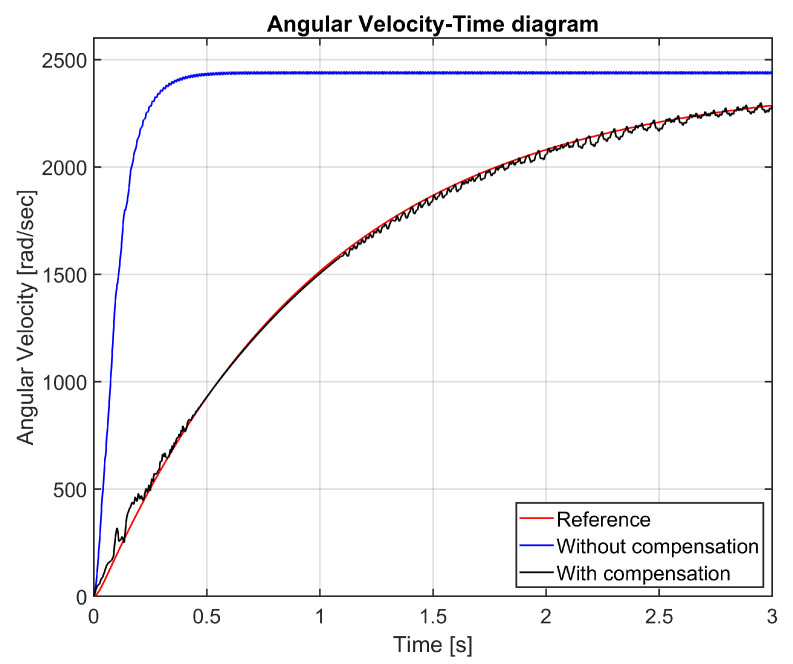
Angular velocity–time diagram.

**Figure 14 sensors-25-00456-f014:**
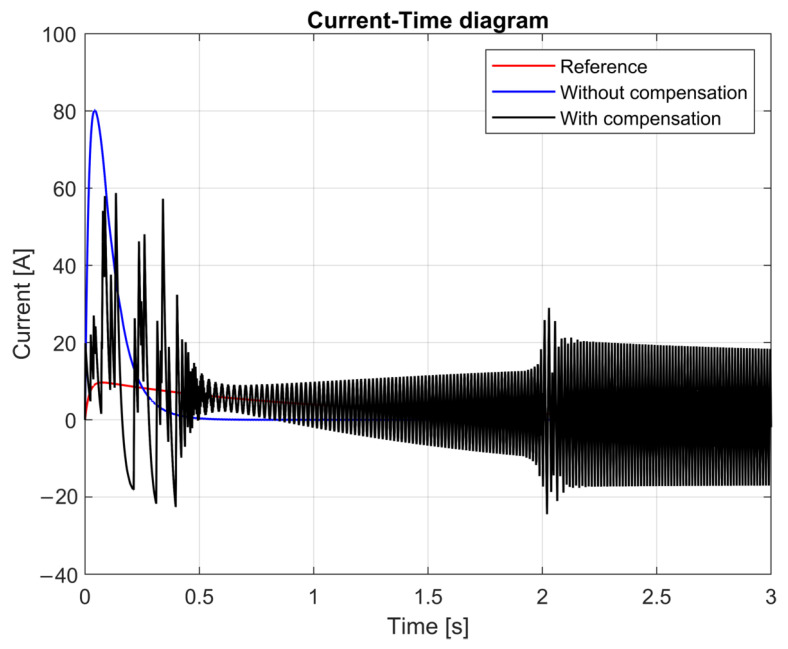
Current–time diagram.

**Figure 15 sensors-25-00456-f015:**
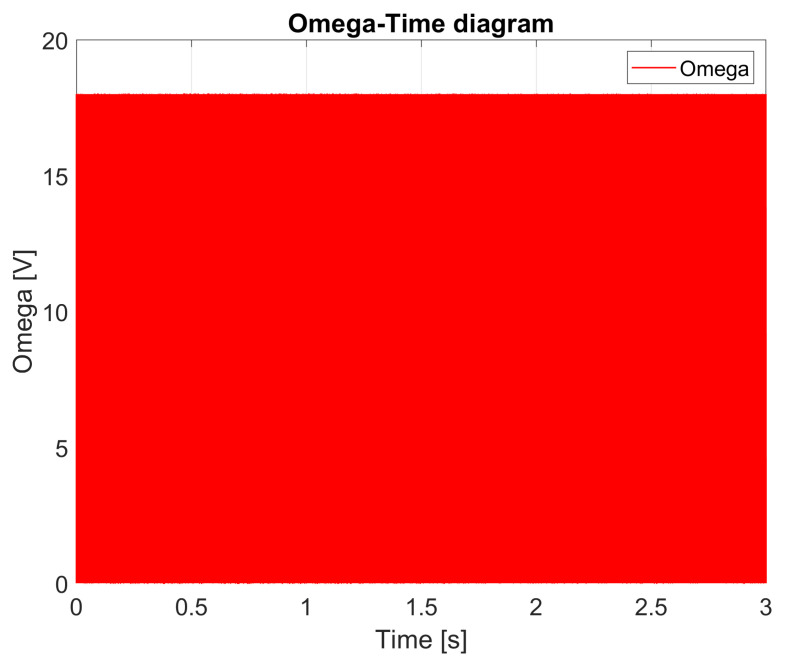
Omega–time diagram.

**Figure 16 sensors-25-00456-f016:**
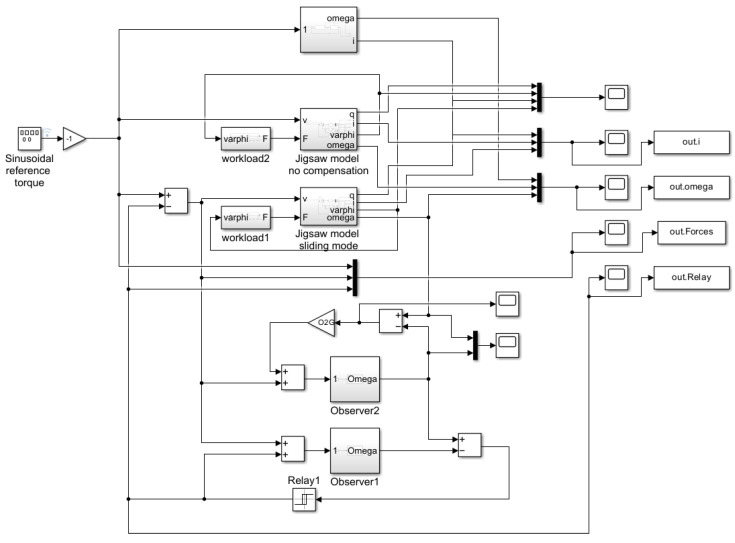
System block diagram with reduced-order state observer.

**Figure 17 sensors-25-00456-f017:**
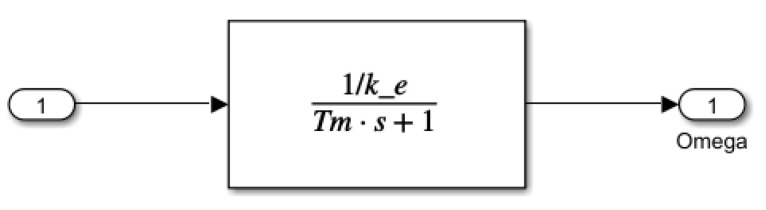
Reduced-order state observer according to (Equation 28).

**Figure 18 sensors-25-00456-f018:**
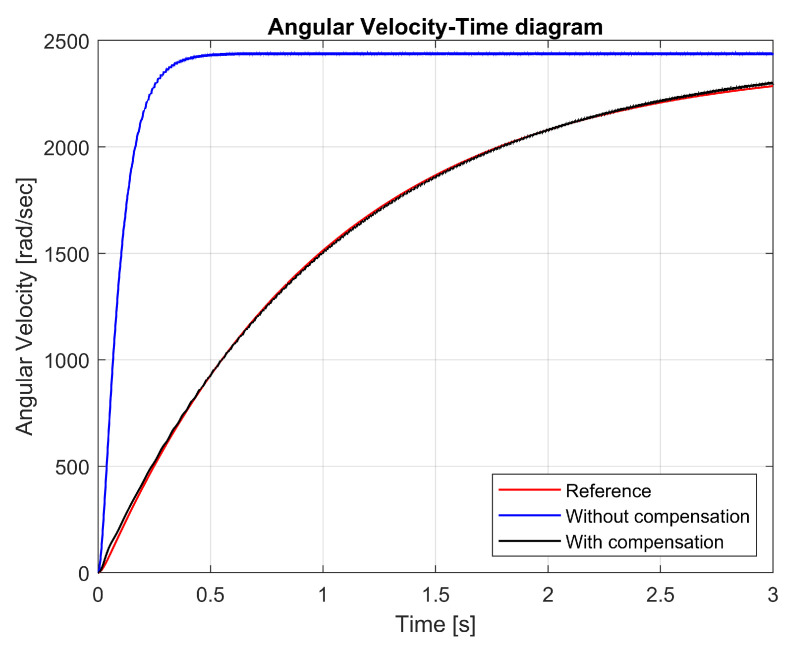
Angular velocity–time diagram with reduced-order state observer.

**Figure 19 sensors-25-00456-f019:**
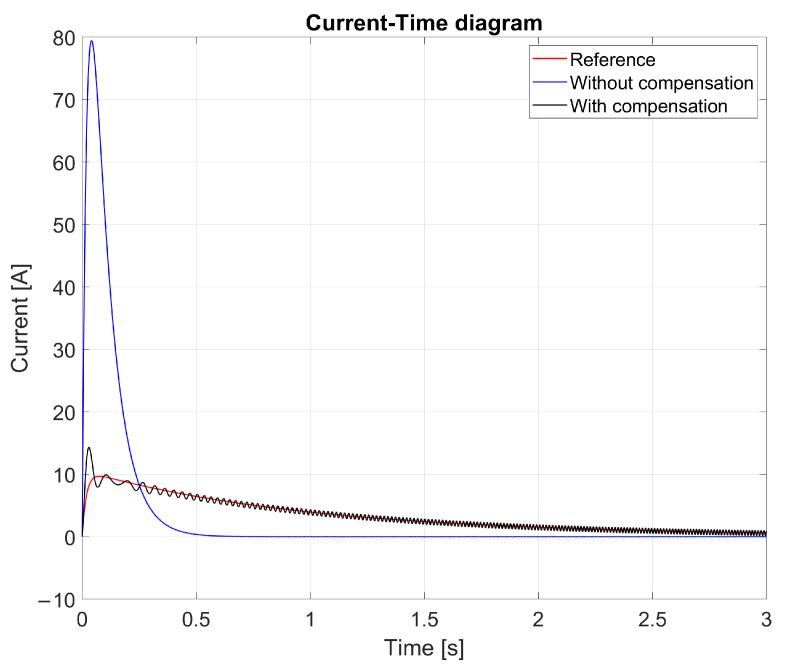
Current–time diagram with reduced-order state observer.

**Figure 20 sensors-25-00456-f020:**
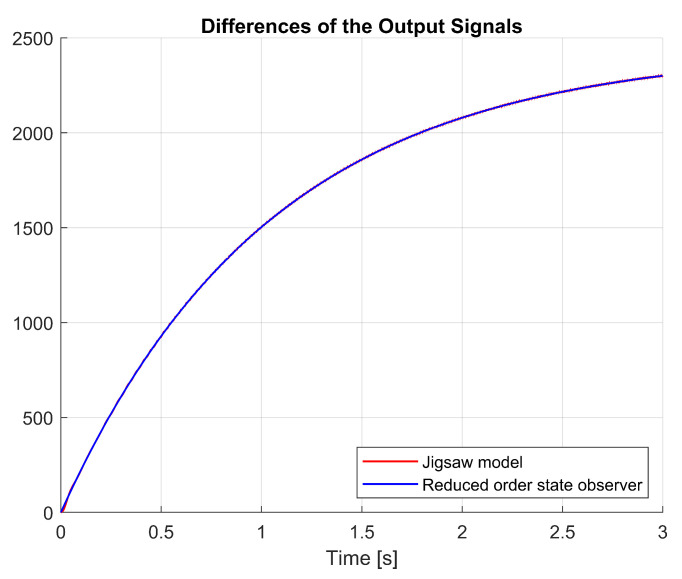
Differences in the output signals.

**Figure 21 sensors-25-00456-f021:**
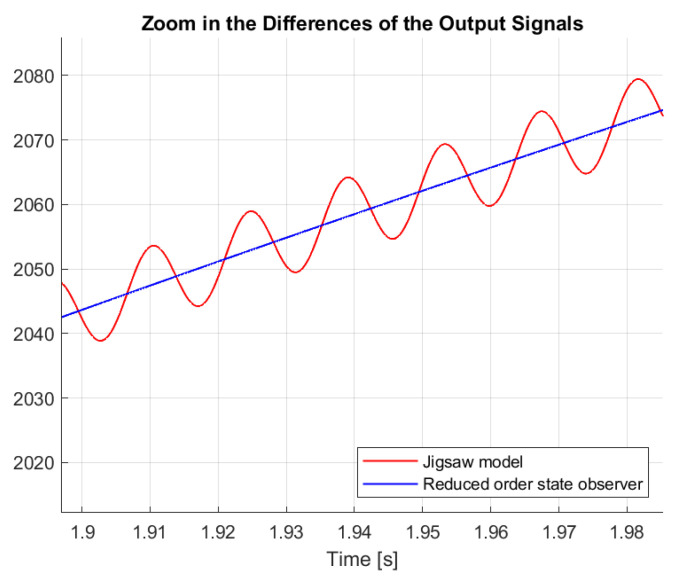
Differences in the output signals.

**Table 1 sensors-25-00456-t001:** Summary of parameters.

Input Parameters	Identified	Unit of Measure
$R_{a}$	0.176	Ω
$R_{b}$	0.0	Ω
$L_{a}$	$3.21\times 10^{-3}$	H
$k_{e}$	$7.383\times 10^{-3}$	V/(rad/s)
$k_{mk}$	6/56	-
$J_{m}$	$24.1\times 10^{-6}$	${\mathrm{kgm}}^{2}$
$k_{m}$	$5.632\times 10^{-3}$	Nm/A
$J_{k}$	$24.0\times 10^{-6}$	${\mathrm{kgm}}^{2}$
$m_{a}$	0.239	kg
$e_{k}$	0.010	m
*b*	$3.4274\times 10^{-6}$	Nms/rad
$v_{in}$	18	V
Γ	18	V
$T_{c}$	0.01	s
$T_{m}$	0.1043	s
$T_{e}$	0.0182	s
${\hat{T}}_{m}$	1.043	s
${\hat{B}}_{m}$	135.4433	rad/Vs
factor	10	-
$O_{2G}$	0.1	1/s
$T_{err}$	0.03	s
$∆a_{42}$	207.3	$1/{\mathrm{As}}^{2}$
$∆a_{44}$	−0.605	1/s
$∆b_{42}$	−0.605	1/kgm
$F_{saw_{max}}$	400	N
$\omega_{max}$	2445	rad/s

## Data Availability

Matlab codes are included in the article, Simulink codes are attached as separate files.

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
