# Peer review of "Investigation of Sliding Mode Control in the Nonlinear Modeling of Cordless Jigsaws"

_sensors, 2025, doi:10.3390/s25020456_

Round 1

Reviewer 1 Report

Comments and Suggestions for Authors

The manuscript systematically describes the use of sliding model controller, the use of nonlinear compensation technology to improve the battery life, to ensure the smooth operation of the motor. The following are some suggestions, specific suggestions are included in the attachment.

1.The manuscript should ensure the accuracy of words and grammar in the process of writing, and the introduction should be more logical and progressive.

2.The drawings in the manuscript suggest further aesthetic treatment.

3.Conclusions should be simplified.

Comments on the Quality of English Language

English grammar needs further improvement and some words in the manuscript need to be used properly.

Author Response

We are grateful for the valuable comments and constructive suggestions. The manuscript has undergone a thorough review and has been revised based on the provided feedback. Significant improvements and clarifications have been made throughout the text, addressing both content and formatting to ensure the manuscript meets the expectations of scientific publication and the provided guidelines. The detailed answers to the questions and suggestions can be found in the attached in the PDF document.

Reviewer 2 Report

Comments and Suggestions for Authors

Strong points

1. This study demonstrates the application and advantages of sliding mode control for a cordless jigsaw, aiming to reduce peak current draw, prolong motor lifespan, and enhance battery efficiency. A key takeaway from this research is the use of model-referenced sliding mode control, leveraging a second-order system model focused on the motor’s angular displacement. By omitting the electrical time constant and current control dynamics, authors implemented a simplified yet effective approach that ensures reliable control without compromising accuracy.

2. 9 quality references of co-authors of the article.

Weak points

1.  Translate the authors' names at the beginning of the article and the text (Lines 331-333) into English.

2. The abstract refers to a  jigsaw, not a handheld stroker.  (Line 2).

3. What's the acronym DIY? (Line 21).

4. Where is the dark arrow in Figure 1?

5. (labled as (6)), not (4) (Line 123). (labled as (4)), not 6). (Line 128).

Author Response

(The authors gave the same response as above.)

Reviewer 3 Report

Comments and Suggestions for Authors

Overall, the work is interesting. However, some comments should be addressed.

1. Background motivation (big background -> small background -> research objectives -> challenges/problems faced in achieving the objectives) -> literature review of related methods (recent rather than early historical) -> further propose problems/challenges to be solved in this paper by analyzing research progress (theoretical or engineering) -> methods of this paper -> contributions -> organization of the full paper.

2. The arrangement of this article is confusing. Why is the "EXPERIMENTAL SYSTEM Description" section placed before the section "Problem statement"? Intuitively, this should be the content of the section "Simulation Setup." The section "Electromechanical Model of the Jigsaw" should belong to the section "Problem statement." In my opinion, the sections "Application of the Theory in Section 3.2" and "Condition for Sliding Mode Stability" belong to the controller design and stability analysis section. Setting up this section and treating them as subsections for easier reading is recommended.

3. Clarify the main contribution of the work and ensure it is clearly emphasized.

4.In lines 123-135, labels (1)(2)(3)(4) contradict the labels in Figure 1, which confuses the readers.

5.Cancel the line break of φ(k) above line 278

6. Address the grammatical errors and typos scattered throughout the paper.

7. Strengthen the discussion of the originality in the introduction section.

8. Provide further explanations for the simulation results presented in the figures to aid reader understanding.

9. In the "Condition for Sliding Mode Stability" section, please use the standard Lyapunov stability theory for analysis.

10. To more strongly verify the effectiveness and characteristics of the proposed method, I suggest that the authors consider comparing their research with the following papers: Overall, the work is interesting. However, some comments should be addressed.
1. Background motivation (big background -> small background -> research objectives ->
challenges/problems faced in achieving the objectives) -> literature review of related methods
(recent rather than early historical) -> further propose problems/challenges to be solved in this
paper by analyzing research progress (theoretical or engineering) -> methods of this paper ->
contributions -> organization of the full paper.
2. The arrangement of this article is confusing. Why is the "EXPERIMENTAL SYSTEM
Description" section placed before the section "Problem statement"? Intuitively, this should be the
content of the section "Simulation Setup." The section "Electromechanical Model of the Jigsaw"
should belong to the section "Problem statement." In my opinion, the sections "Application of the
Theory in Section 3.2" and "Condition for Sliding Mode Stability" belong to the controller design
and stability analysis section. Setting up this section and treating them as subsections for easier
reading is recommended.
3. Clarify the main contribution of the work and ensure it is clearly emphasized.
4.In lines 123-135, labels (1)(2)(3)(4) contradict the labels in Figure 1, which confuses the
readers.
5.Cancel the line break of φ(k) above line 278
6. Address the grammatical errors and typos scattered throughout the paper.
7. Strengthen the discussion of the originality in the introduction section.
8. Provide further explanations for the simulation results presented in the figures to aid reader
understanding.
9. In the "Condition for Sliding Mode Stability" section, please use the standard Lyapunov
stability theory for analysis.
10. To more strongly verify the effectiveness and characteristics of the proposed method, I suggest that the authors consider comparing their research with the following papers:
https://doi.org/10.1007/s11071-023-08881-1
, or cite them in the research progress in the introduction to highlight the advanced and advantages of the proposed method.

11. The problem solved and the advantages of the proposed method described in the conclusion should be clearly described in the introduction and simulation section in advance. There is no need to elaborate in detail in the conclusion. It is recommended to use a paragraph (including future research plans) to clearly and highlight the conclusion, referring to the format of 10.1109/TAC.2018.2794885.

Comments on the Quality of English Language

Need to be improved.

Author Response

(The authors gave the same response as above.)

Reviewer 4 Report

Comments and Suggestions for Authors

The file is uploaded

Comments on the Quality of English Language

Can be Improved

Author Response

(The authors gave the same response as above.)

Round 2

Reviewer 3 Report

Comments and Suggestions for Authors

My questions have been adequately addressed. Then I think that the manuscript can be published.